# Use of insecticide treated nets in children under five and children of school age in Nigeria: Evidence from a secondary data analysis of demographic health survey

Chinazo N. Ujuju[1]*, Chukwu Okoronkwo[2], Okefu Oyale Okoko[2], Adekunle Akerele[3], Chibundo N. Okorie[4], Samson Babatunde Adebayo[5]

1 Data for Decisions Nigeria Ltd, Abuja, Nigeria, 2 National Malaria Elimination Programme (NMEP), Federal Ministry of Health, Abuja, Nigeria, 3 Department of Medical Statistics and Epidemiology, University of Ibadan, Ibadan, Nigeria, 4 Department of Pharmaceutical Microbiology and Biotechnology, Faculty of Pharmaceutical Sciences, University of Nigeria, Nsukka, Enugu, Nigeria, 5 National Agency for Food and Drug Administration & Control, Lagos, Nigeria

* Chinazoujuju@gmail.com

**Data Availability Statement:** The data underlying the results presented in this study are available on

## Abstract

### Background and objective

Use of insecticide treated nets (ITN), one of the most cost-effective malaria interventions contributes to malaria cases averted and reduction in child mortality. We explored the use of ITN in children under five (CU5) and children of school age to understand factors contributing to ITN use.

### Methods

A cross-sectional study analyzed 2018 Nigeria Demographic and Health Survey data. The outcome variable was CU5 or children of school age who slept under ITN the night before the survey. Independen*t* variables include child sex, head of household's sex, place of residence, state, household owning radio and television, number of household members, wealth quintile, years since ITN was obtained and level of malaria endemicity. Multi-level logistic regression model was used to access factors associated with ITN use among children.

### Results

In total, 32,087 CU5 and 54,692 children of school age were examined with 74.3% of CU5 and 57.8% of children of school age using ITN the night before the survey. While seven states had more than 80% of CU5 who used ITN, only one state had over 80% of school children who used ITN. ITN use in CU5 is associated with living in rural area (aOR = 1.20, 95% CI 1.14 to 1.26) and residing in meso endemic area (aOR = 3.1, 95% CI 2.89 to 3.54). While In children of school age, use of ITN was associated with female headed households (aOR = 1.14, 95% CI 1.09 to 1.19), meso (aOR = 3.17, 95% CI 2.89 to 3.47) and hyper (aOR =

DHS program website https://www.dhsprogram.com/data/.

**Funding:** The authors received no funding for this work.

**Competing interests:** The authors have declared that no competing interest exist.

14.9, 95% CI 12.99 to 17.07) endemic areas. Children residing in larger households were less likely to use ITN.

## Conclusions

This study demonstrated increased use of ITN in CU5 from poor households and children living in rural and malaria endemic areas. Findings provide some policy recommendations for increasing ITN use in school children.

## Introduction

Malaria remains a global public health problem with 229 million malaria cases and 409 malaria related death reported globally in 2019 [1]. Sub-Saharan African countries contribute 85% of the malaria cases globally with Nigeria accounting for 27% of the global malaria cases and 23% of global malaria deaths [1]. Nigeria, home of over 200 million people, has the majority of her population at high risk of malaria. Prevalence of malaria in children under five (CU5) is more than 50% in Kebbi state while Lagos, Imo and Anambra state have prevalence less than 10%. The remaining 32 states and the Federal Capital Territory have prevalence between 11% and 50% [2]. The prevalence of malaria among CU5 in Nigeria decreased from 27% in 2015 to 23% in 2018 [2]. Recent studies in sub-Saharan Africa revealed an increased malaria parasitaemia in school children [3–5]. Some published and unpublished studies in Nigeria have reported increased prevalence of malaria in children of school age. For instance, a study conducted in Bayelsa state reported higher prevalence of malaria among children 6–8 years old [6]. Another study conducted in Plateau and Abia states reported high prevalence among children 5–9 years [7]. While malaria in children of school age is not associated with severity, absenteeism from school and anaemia is common among this age group [8, 9].

Several interventions have been deployed globally to tackle morbidity and mortality due to malaria. These include prompt diagnostic tests to confirm malaria prior to treatment with artemisinin-based combination therapy (ACT), prevention of malaria in pregnant women, seasonal malaria chemoprevention in children 3–59 months and vector control using insecticide treated nets and in-door residual spraying. Vector control is one of the most important approaches for eradicating malaria as it aims to interrupt malaria transmission. While all these aforementioned interventions have been found to be effective, implementation has substantial cost implications [10, 11]. Consequently, insecticide treated net (ITN) use has been identified as the most cost-effective malaria intervention and has largely contributed to over 50% of malaria cases averted and reduction of child mortality by 27% [12, 13].

To harness the benefits of ITNs, increase population access to ITN and rapidly achieve universal coverage, mass ITN distribution campaigns have been implemented in Nigeria [14–16]. ITN campaigns conducted once every three years are implemented with a lot of SBCC messages and these messages were effective in improving net culture and use especially for vulnerable groups [16]. Keep-up channels using continuous distribution mechanism maintain coverage between ITN campaigns. Children under five and pregnant women who are most vulnerable to malaria are prioritized through routine net distribution channels during antenatal care services and immunization clinics [17–19].

The Nigeria National Malaria Strategic Plan (NMSP) aims at improving access and utilization of vector control interventions to 80% of the target population by 2025. Use of ITN by CU5 living in a household with at least one ITN increased from 58.6% in 2010 to 74.3% in

2018. However, the use of ITN by children of school age increased from 37.8% to 57.6% [2] with children of school age having the lowest proportion of ITN use in 2018 compared to other age group [2]. The low use of ITN by children of school age compared to other age group in the family has also been documented by Olapeju et al. [20].

Most studies on use of ITNs focus on general population with more emphasis on pregnant women and CU5 [21–25]. No study in Nigeria has been identified that explored the use of ITN in all children including children of school age and the progress made in achieving the ITN utilization target of 80% set by the NMSP at state level using a national survey. This study aims to conduct an analysis of ITN use in CU5 and children of school age at national and sub-national level with the aim of understanding ITN use for these children. Understanding factors associated with ITN use in children is timely to inform future intervention among this target group.

## Methods

This cross-sectional study analyzed the 2018 Nigeria Demographic and Health Survey (NDHS) dataset. NDHS is a nationally representative survey with samples drawn from all states and Local Government Areas (LGAs) based on the sampling frame of enumeration areas in the country. Methods for sampling and fieldwork are described in the NDHS survey report. This study analyzed merged persons recode (PR) file and the household recode (HR) filtered for household identification number, any ITN in household and number of ITN in household. Data were adjusted for survey design clustering and non-response by applying the individual weight provided in the NDHS dataset to every analysis.

### Target population

Target population for the analysis was CU5 (aged 0–4 years) and children of school age (aged 5–14 years) who slept under ITN the night before the survey in households with at least one ITN.

### ITN campaign in Nigeria

In Nigeria, ITN campaign was conducted for the first time in 2009 and has been implemented on a rolling basis since then. In 2015, ITN campaign was conducted in Abia, Cross River, Ebonyi, Kano and Kaduna states. In 2016, only Benue and Oyo states conducted an ITN campaign. Adamawa, Edo, Imo, Kogi, Kwara, Ondo and Osun conducted ITN campaign in 2017 and Sokoto, Bauchi, Gombe, Jigawa, Katsina, Nasarawa, Ogun and Akwa Ibom states conducted ITN campaign in 2018. The following states did not conduct ITN campaign between 2015 and 2018: Zamfara, FCT, Niger, Yobe, Borno, Kebbi, Plateau, Taraba, Ekiti, Anambra Enugu, Rivers, Bayelsa, Delta and Lagos states.

### Variables

**Outcome variables.** The outcome variable is ITN use in CU5 and children of school age. Use of ITN was defined as whether a child under five or 5–14 years living in a household that owns at least one ITN slept under an ITN the night before the survey.

**Independent variables.** The dataset was examined for variables of interest that were likely to influence the utilisation of ITNs. Literatures were also considered in identifying factors that could influence the utilization of ITN in children [20, 26, 27]. The independent variables considered for analysis were child sex, household characteristics such as head of household's sex, household ownership of radio and television, number of household members, when ITN was

obtained and wealth index. Demographic characteristics such as place of residence, state and region as well as malaria endemicity were included in the analysis. Malaria endemicity was classified into hypo-endemicity (states with prevalence less than 10%), meso-endemicity (10–50%) and hyperendemicity (51–75%) using state prevalence of malaria in CU5 obtained from 2018 NDHS. A study using *Plasmodium falciparum* parasite rate (*Pf*PR) provided a basis for the classical categorical measures of malaria transmission into hypo-endemic (<10%), meso-endemic (10–50%), and hyper-endemic (51–75%) and this measure has been used in previous studies [28].

## Statistical analysis

Statistical analysis was conducted with STATA version 14 and three levels of analysis conducted. Firstly, distribution of variables was conducted using frequency and proportion. Bivariate analysis was subsequently conducted to determine the level of association between the outcome variable, use of ITN in CU5 or children of school age with the independent variables with significant measure at p<0.05. Variable Inflation Factor (VIF) was calculated to determine the extent of the multi-collinearity of the independent variables and their suitability to be included into the multilevel analysis [29]. Variables with p-value <0.2 at bivariate level were included into a multi-level logistics regression model used to assess factors influencing the use of ITN in CU5 and children of school age. The nested structure of the demographic health survey (DHS) data in which children were selected from household within communities necessitated the use of the methodology. Use of multilevel logistic regression models for the analysis of DHS data has been documented and used severally in literature [29–31] therefore we would not document the theory in this paper. For this paper we constructed a model for CU5 and children of school age independently. Three models were constructed for each category which included the household model, the community level model and the combined model. The outcome variables for each model were use of ITN with "1" for use and "0" for non-use of ITN. We reported the variance and standard deviation at the household and community levels for each model, the residual, the log likelihood, the intraclass correlation, the Akaike information criteria and the Bayesian information criteria. Variables found to be correlated with other variables would be exempted from the logistic regression analysis. Significance was assessed based on 95% confidence interval of odds ratio not including 1

## Ethical consideration

This work examined a population-based dataset accessed online from The Demographic Health Survey (DHS) Program. The DHS Program adheres to guidelines for protecting the privacy of respondents by removing all personal identifiers. As The DHS Program sought and received ethical approval before the survey, this research did not require any additional ethical approvals. However, The DHS Program granted permission to use the dataset for this work.

## Results

### Univariate

Data on 86,778 children were analyzed with 32,087 CU5 and 54,692 children of school age. While 70% (n = 22,440) of CU5 live in households with at least one ITN, 68.6% (n = 37,502) of children of school age live in households with at least one ITN. Table 1 shows the demographic characteristics of CU5 and children of school age. Half (50.8%) of the children were male and 88.9% of the head of households were male. The majority of households (59.9%) have a radio while 44.6% indicated that the household owns a television. About 43% of nets were obtained

**Table 1. Demographic characteristics of CU5 (0–4) years and children of school age (5–14 years).**

| Variables | Children under five | Children of school age | Total |
|---|---|---|---|
| | n (%) | n (%) | n (%) |
| **Sex** | | | |
| Male | 16,366 (51.0) | 27,709 (50.7) | 44,075 (50.8) |
| Female | 15,721 (49.0) | 26,983 (49.3) | 42,704 (49.2) |
| **Sex of head of household** | | | |
| Male | 29,117 (90.8) | 48,027 (87.8) | 77,114 (88.9) |
| Female | 2,969 (9.2) | 6,665 (12.2) | 86,778 (11.1) |
| **Household own radio** | | | |
| No | 13,361 (41.6) | 21,469 (39.3) | 34,831 (40.1) |
| Yes | 18,726 (58.4) | 33,222 (60.7) | 51,948 (59.9) |
| **Household own TV** | | | |
| No | 17,760(55.4) | 30,288 (55.4) | 48,047 (55.4) |
| **Yes** | 14,327 (44.6) | 24,404 (44.6) | 38,731 (44.6) |
| **When ITN was obtained** | | | |
| Less than one year | 5,705 (34.0) | 7,811 (35.8) | 13,516 (35.1) |
| 1–3 years | 7,507 (44.8) | 9,101 (41.7) | 16 609 (43.0) |
| More than 3 years | 3,550 (21.2) | 4,931 (22.6) | 8,481 (22.0) |
| **Number of Household members** | | | |
| 1–3 persons | 3,202 (10.0) | 2,728 (5.0) | 5,930 (6.8) |
| 4–6 persons | 14,442 (45.0) | 20,750 (37.9) | 35,192 (40.6) |
| 7–9 persons | 7,832 (24.4) | 16,965 (31.0) | 24,797 (28.6) |
| >9 persons | 6,611 (20.6) | 14,248 (26.1) | 20,859 (24.0) |
| **Wealth quintiles** | | | |
| Poorest | 6,988 (21.8) | 12,308 (22.5) | 19,296 (22.2) |
| Poorer | 7,109 (22.2) | 11,574 (21.2) | 18,682 (21.5) |
| Middle | 6,587 (20.5) | 11,041 (20.2) | 17,628 (20.3) |
| Richer | 5,948 (18.5) | 10,370 (19.0) | 16,318 (18.8) |
| Richest | 5,456 (17.0) | 9,398 (17.2) | 14,854 (17.1) |
| **Residence** | | | |
| Urban | 12,638 (39.4) | 22,439 (41.0) | 35,077 (40.4) |
| Rural | 19,448 (60.6) | 32,253 (59.0) | 51,701 (59.6) |
| **Region** | | | |
| North Central | 4,371 (13.6) | 7,112 (13.0) | 11,483 (13.2) |
| North East | 5,885 (18.3) | 10,483 (19.2) | 16,368 (18.9) |
| North West | 11,246 (35.1) | 19,088 (34.9) | 30,334 (35.0) |
| South East | 3,393 (10.6) | 5,220 (9.6) | 8,613 (9.9) |
| South South | 2,915 (9.1) | 5,289 (9.7) | 8,204 (9.5) |
| South West | 4,276 (13.3) | 7,500 (13.7) | 11,776 (13.6) |
| **States** | | | |
| Abia | 426 (1.3) | 607 (1.1) | 1,033 (1.2) |
| Cross river | 304 (1.0) | 595 (1.1) | 899 (1.0) |
| Ebonyi | 824 (2.6) | 1,287 (2.4) | 2,111 (2.4) |
| Kano | 2,471 (7.7) | 4,463 (8.2) | 6,934 (8.0) |
| Kaduna | 2,090 (6.5) | 3,349 (5.9) | 5,339 (6.2) |
| Benue | 921 (2.9) | 1,276 (2.3) | 2,197 (2.5) |
| Oyo | 944 (2.9) | 1,567 (2.9) | 2,512 (2.9) |
| Adamawa | 757 (2.4) | 1,181 (2.2) | 1,938 (2.2) |

(*Continued*)

**Table 1.** (Continued)

| Variables | Children under five | Children of school age | Total |
|---|---|---|---|
| | n (%) | n (%) | n (%) |
| Edo | 409 (1.3) | 786 (1.4) | 1,195 (1.4) |
| Imo | 652 (2.0) | 1,010 (1.9) | 1,662 (1.9) |
| Kogi | 380 (1.2) | 681 (1.3) | 1,061 (1.2) |
| Kwara | 518 (1.6) | 983 (1.8) | 1,501 (1.7) |
| Ondo | 392 (1.2) | 751 (1.4) | 1,142 (1.3) |
| Osun | 565 (1.8) | 962 (1.8) | 1,526 (1.8) |
| Sokoto | 921 (2.9) | 1,514 (2.8) | 2,435 (2.8) |
| Bauchi | 1,383 (4.3) | 2,445 (4.5) | 3,829 (4.4) |
| Gombe | 647 (2.0) | 1,167 (2.1) | 1,814 (2.1) |
| Jigawa | 1,319 (4.1) | 2,357 (4.3) | 3,677 (4.34) |
| Katsina | 2,203 (6.9) | 3,737 (6.8) | 5,940 (6.8) |
| Nasarawa | 494 (1.5) | 798 (1.5) | 1,292 (1.5) |
| Ogun | 606 (1.9) | 1,015 (1.9) | 1,621 (1.9) |
| Akwa Ibom | 517 (1.6) | 950 (1.7) | 1,467 (1.6) |
| Zamfara | 1,209 (3.8) | 2,053 (3.8) | 3,262 (3.8) |
| FCT Abuja | 217 (0.7 | 349 (0.6) | 566 (0.7) |
| Niger | 1,226 (3.8) | 1,899 (3.5) | 3,125 (3.6) |
| Yobe | 1,210 (3.8) | 2,327 (4.3) | 3,537 (4.1) |
| Borno | 1,146 (3.6) | 2,155 (3.9) | 3,300 (3.8) |
| Kebbi | 1,034 (3.2) | 1,715 (3.1) | 2,749 (3.2) |
| Plateau | 615 (1.9) | 1,125 (2.1) | 1,740 (2.0) |
| Taraba | 743 (2.3) | 1,208 (2.2) | 1,950 (2.3) |
| Ekiti | 304 (1.0) | 513 (0.9) | 817 (0.9) |
| Anambra | 1,029 (3.2) | 1,436 (2.6) | 2,464 (2.8) |
| Enugu | 462 (1.4) | 880 (1.6) | 1,342 (1.6) |
| Rivers | 877 (2.7) | 1,437 (2.6) | 2,314 (2.7) |
| Bayelsa | 224 (0.7) | 422 (0.8) | 1,682 (0.7) |
| Delta | 584 (1.8) | 1,099 (2.0) | 1,683 (1.9) |
| Lagos | 1,466 (4.6) | 2,692 (4.9) | 4, 158 (4.8) |
| **Malaria endemicity** | | | |
| Hypoendemic | 3,147 (9.8.0) | 5,139 (9.4) | 8,285 (9.5) |
| Mesoendemic | 27,906 (87) | 47,838 (87.5) | 75,744 (87.3) |
| Hyperendemic | 1,034 (3.2) | 1,715 (3.1) | 2,749 (3.2) |
| **Ownership of ITN** | | | |
| No ITN | 9,647(30) | 17,190(31.4) | 26,837(30.9) |
| At least 1 ITN | 22,440(70) | 37,502(68.6) | 59,942(69.1) |
| **Total** | **32,087 (100)** | **54,692 (100)** | **86,778 (100)** |

within the past 1–3 years with more CU5 (44.8%) obtaining their net within 1–3 years ago. About forty-one percent (40.6%) of children live in households with 4–6 persons. More CU5 live in households with 4–6 person (45%) compared to children of school age. About 60% of all children included in the analysis reside in rural areas, while 87% reside in meso-endemic area with prevalence between 11% and 50%.

## Bivariate

Table 2 presents the results of the bivariate analyses of ITN use in CU5 and children of school age by demographic characteristics. Findings among CU5 show that out of 22,440 children that live in households with at least one ITN, 74.3% (n = 16,671) slept under ITN the night before the survey. ITN use was associated with ownership of radio, television, when ITN was obtained, number of household members, wealth quintiles, place of residence, region, states and malaria endemicity ($p < 0.05$). Children under five from households with less than 6 members were most likely to sleep under an ITN. Similarly, CU5 who obtained ITN less than 3 years (99.7%) were also more likely to sleep under ITN.

A higher proportion of CU5 (77.9%) from the poorest wealth quintiles slept under ITN compared with those from the highest wealth quintile (67%). Use of ITN was significantly associated (p<0.001) with place of residence. About 75.8% of CU5 from rural areas slept under ITN compared to 71% from urban areas. A substantial geographical variation was noticed on use of ITN with CU5 from North West region being most likely to sleep under an ITN (80.3%) while in South-South, 63.1% of CU5 slept under ITN. Under five children living in malaria hypo-endemic area were least likely to sleep under ITN (56.1%) compared with 95% of under five children in hyperendemic areas. Considering the states with up to 80% ITN utilization in CU5, Ebonyi state (89.1%), Kano states (82.3%), Benue state (93.1%), Adamawa (90.0%) Kebbi (95.0%) Plateau (85.6%) and Jigawa state (90.9%) had above 80% ITN utilization in CU5.

Turning attention to findings on children of school age, out of 37,502 children of school age that live in households with least one ITN, 57.8% (n = 21,690) slept under ITN the night before the survey. Similar significant associations as in the case of CU5 were observed. While no differentials of child's sex, head of household sex and ownership of television were evident in the case of CU5, these variables were significantly associated with ITN use among children of school age. Furthermore, ITN use is significantly associated with number of household members, wealth quintiles, region, state and malaria endemicity. Considering the states with up to 80% ITN utilization in children of school age, only Jigawa state has over 80 percent of children of school age that slept under ITN the night before the survey. Table 3 with crude odds ratio (COR) of factors associated with ITN use in CU5 and children of school age is included in S1 File.

## Multivariate results

The VIF computation done before fitting the multilevel logistic regression models for under five and above five revealed a mean VIF score of 3.25 and 3.91 for CU5 and children of school age respectively after removing the variable state and region due to collinearity.

Table 3 presents the results of the multilevel logistic regress with three models for each of CU5 and school age children. For the CU5, the household model demonstrated that children from richer (aOR = 0.25, 95% CI 0.09 to 0.65) and richest (aOR = 0.18, 95% CI 0.06 to 0.53) wealth quintile were less likely to utilize ITN, the community model demonstrated that households in rural area (aOR = 1.20, 95% CI 1.14 to 1.26) and in meso endemic (aOR = 3.10, 95% CI 2.89 to 3.54) areas were more likely to use ITN. The combined model for CU5 demonstrated that having TV (aOR = 2.14, 95% CI 1.02 to 4.50) and living in rural areas (aOR = 2.93, 95% CI 1.14 to 1.26) contributed to using ITN while households with 4–6 persons (aOR = 0.37, 95% CI 0.16 to 0.89), 7 to 9 persons (aOR = 0.34, 95% CI 0.14 to 0.85), and >9 persons (aOR = 0.35, 95% CI 0.14 to 0.90) and households from richest quintiles (aOR = 0.19, 95% CI 0.04 to 0.78) were less likely to use ITN. Variability in use of ITN was highest in the household with an intraclass correlation of 14%.

**Table 2. Use of ITN in CU5 and children of school age living in households with at least one ITN by demographic characteristics.**

| Variable | Children under five | | Children of school age | |
|---|---|---|---|---|
| | Slept under ITN (n = 16,671) | P Value | Slept under ITN n = 21,690 | P Value |
| **Sex** | n (%) | <0.417 | n (%) | <0.001 |
| Male | 8,353 (74.6) | | 10,582 (55.9) | |
| Female | 8,148 (74.0) | | 11,109 (59.8) | |
| **Sex of head of household** | | 0.712 | | 0.001 |
| Male | 15,277 (74.2) | | 19,158 (57.3) | |
| Female | 1,394 (74.8) | | 2,532 (62.2) | |
| **Household own radio** | | <0.001 | | 0.267 |
| No | 7,316 (76.7) | | 8,800 (58.5) | |
| Yes | 9,355 (72.5) | | 12,890 (57.4) | |
| **Household own TV** | | <0.001 | | 0.01 |
| No | 10,268 (77.5) | | 13,248 (59.1) | |
| Yes | 6,404 (69.7) | | 8,442 (56.0) | |
| **When ITN was obtained** | | <0.046 | | <0.665 |
| Less than one year | 18 (0.3) | | 7,708 (99.6) | |
| 1–3 years | 21 (0.3) | | 8,965 (99.5) | |
| More than 3 years | 27 (0.8) | | 4,819 (99.5) | |
| **Number of Household members** | | <0.001 | | <0.001 |
| 1–3 persons | 1,755 (85.8) | | 1,018 (69.3) | |
| 4–6 persons | 7,608(78.0) | | 8,630(64.8) | |
| 7–9 persons | 3,980 (71.3) | | 6,891 (57.9) | |
| >9 persons | 3,327 (65.7) | | 5,151 (47.6) | |
| **Wealth quintiles** | | <0.001 | | <0.021 |
| Poorest | 4,165 (77.9) | | 5,507 (58.0) | |
| Poorer | 4,179 (76.8) | | 5.219 (60.1) | |
| Middle | 3,525 (75.4) | | 4,487 (59.0) | |
| Richer | 2,667 (70.1) | | 3,639 (54.9) | |
| Richest | 2,136 (67.3) | | 2,838 (55.8) | |
| **Residence** | | <0.001 | | <0.116 |
| Urban | 5,640 (71.5) | | 7,766 (56.6) | |
| Rural | 11,032 (75.8) | | 13,925 (58.6) | |
| **Region** | | <0.001 | | <0.001 |
| North Central | 2,057 (76.2) | | 2,314 (56.5) | |
| North East | 2,777 (69.7) | | 3,648 (51.8) | |
| North West | 8,138 (80.3) | | 10,644 (62.3) | |
| South East | 1,234 (66.2) | | 1,493 (54.4) | |
| South South | 975 (63.1) | | 1,447 (52.9) | |
| South West | 1,490 (67.5) | | 2,146 (56.3) | |
| | | <0.001 | | <0.001 |
| **States** | | | | |
| Abia | 101 (47.7) | | 134 (45.3) | |
| Cross River | 130 (71.6) | | 234 (62.5) | |
| Ebonyi | 556 (89.1) | | 777 (79.4) | |
| Kano | 1,827 (82.3) | | 2,579 (65.9) | |
| Kaduna | 1,386 (78.0) | | 1,667 (61.4) | |
| Benue | 595 (93.1) | | 641 (76.2) | |

*(Continued)*

**Table 2.** (Continued)

| Variable | Children under five | | Children of school age | |
|---|---|---|---|---|
| | Slept under ITN | P Value | Slept under ITN | P Value |
| | (n = 16,671) | | n = 21,690 | |
| Oyo | 389 (76.9) | | 597 (73.6) | |
| Adamawa | 360 (90.0) | | 478 (75.5) | |
| Edo | 126 (53.1) | | 182 (38.9) | |
| Imo | 196 (51.6) | | 199 (33.7) | |
| Kogi | 205 (71.0) | | 291 (58.8) | |
| Kwara | 172 (50.8) | | 238 (36.7) | |
| Ondo | 232 (69.8) | | 340 (60.7) | |
| Osun | 179 (62.9) | | 265 (49.7) | |
| Sokoto | 516 (64.1) | | 529 (40.5) | |
| Bauchi | 723 (60.8) | | 803 (38.7) | |
| Gombe | 246 (52.6) | | 303 (34.1) | |
| Jigawa | 1,168 (90.9) | | 1,941 (83.9) | |
| Katsina | 1,597 (77.5) | | 2,266 (65.0) | |
| Nasarawa | 281 (70.9) | | 338 (58.9) | |
| Ogun | 296(78.4) | | 404 (67.5) | |
| Akwa Ibom | 189 (52.4) | | 236 (37.7) | |
| Zamfara | 678 (69.7) | | 395 (23.8) | |
| Yobe | 712 (78.1) | | 1,080 (63.1) | |
| Borno | 559 (77.2) | | 792 (62.2) | |
| Kebbi | 967 (95.0) | | 1,268 (75.3) | |
| Niger | 450 (74.9) | | 391 (44.7) | |
| FCT Abuja | 76 (68.4) | | 67 (46.7) | |
| Plateau | 278 (85.6) | | 349 (66.4) | |
| Taraba | 178 (60.4) | | 192 (42.2) | |
| Ekiti | 79 (52.9) | | 112 (45.2) | |
| Anambra | 259 (59.5) | | 234 (46.0) | |
| Enugu | 123 (56.9) | | 147 (40.1) | |
| Rivers | 243 (65.1) | | 336 (59.8) | |
| Bayelsa | 71 (69.7) | | 117 (59.3) | |
| Delta | 216 (74.5) | | 341 (67.6) | |
| Lagos | 315 (56.5) | | 369 (38.4) | |
| **Malaria endemicity** | | <0.001 | | <0.001 |
| Hypoendemic | 770 (56.1) | | 802 (38.9) | |
| Mesoendemic | 14,935 (74.5) | | 19,620 (58.1) | |
| Hyperendemic | 967 (95.0) | | 1,268 (75.3) | |
| Total | 16,671(74.3) | | 21,690 (57.8%) | |

For the school age children's models, the household model demonstrated that children from household with female head (aOR = 1.14, 95% CI 1.09 to 1.19) and having a television (aOR = 1.09, 95% CI 1.01 to 1.16) were likely to use ITN while children living in household with 7 to 9 persons (aOR = 0.67, 95% CI 0.60 to 0.74) or >9 persons (aOR = 0.39, 95% CI 0.36 to 0.44) were less likely to use ITN. The community model demonstrated that children living in rural (aOR = 1.08, 95% CI 1.04 to 1.12), coming from meso endemic (aOR = 3.17, 95% CI 2.89 to 3.47) and hyper endemic (aOR = 14.9, 95% CI 12.99 to 17.07) areas were more likely to

**Table 3. Multilevel logistic regression on socio and demographic factors associated with ITN use in CU5 and children of school age in households owning at least one ITN.**

| Fixed Effect | | | | | | | |
|---|---|---|---|---|---|---|---|
| Characteristics | Categories | CU5 aOR (95% CI) | | | Children of school age aOR (95% CI) | | |
| | | Household | Community | Combined | Household | Community | Combined |
| Sex of head of household | Male | | | | | | |
| | Female | | | | 1.14 (0.89,2.17) | | 1.14(1.09,1.19) |
| Household own radio | No | | | | | | |
| | Yes | 1.29 (0.87,1.92) | | | | | |
| Household own TV | No | | | | | | |
| | Yes | 1.82 (0.95,3.50) | | 2.14(1.02,4.50) | 1.09(1.02,4.5) | | 1.08(1.00,1.15) |
| When net was obtained | Less than one year1 | | | | | | |
| | 1–3 years | 0.78(0.5,1.23) | | 0.76(0.45,1.27) | | | |
| | More than 3 years | 1.03 (0.58,1.85) | | 0.95(0.50,1.81) | | | |
| Number of Household members | 1 to 3 persons | | | | | | |
| | 4 to 6 persons | 1.05 (0.59,1.86) | | 0.37(0.16,0.89) | 0.93 (0.46,1.27) | | 0.92(0.84,1.02) |
| | 7 to 9 persons | 1.29 (0.67,2.48) | | 0.34(0.14,0.85) | 0.67(0.5,1.82) | | 0.66(0.59,0.72) |
| | >9 persons | 1.06 (0.54,2.07) | | 0.35(0.14,0.90) | 0.4(0.16,0.88) | | 0.38(0.35,0.43) |
| Socio Economic Status | Poorest | | | | | | |
| | Poorer | 0.71 (0.38,1.32) | | 0.61(0.30,1.25) | 1.01 (0.14,0.86) | | 1.03(0.96,1.11) |
| | Middle | 1.22 (0.49,3.01) | | 1.49(0.49,4.51) | 1.02(0.14,0.9) | | 1.08(0.99,1.19) |
| | Richer | 0.25 (0.09,0.65) | | 0.3(0.08,1.06) | 1.05(0.3,1.25) | | 1.15(1.02,1.29) |
| | Richest | 0.18 (0.06,0.53) | | 0.19(0.04,0.78) | 1.14 (0.49,4.52) | | 1.30(1.14,1.49) |
| Residence | Urban | | | | | | |
| | Rural | | 1.20 (1.14,1.26) | 2.93 (0.84,10.24) | | 1.08 (1.04,1.12) | 1.37(1.19,1.59) |
| Malaria endemicity | Hypo endemic | | | | | | |
| | Meso endemic | | 3.10 (2.89,3.54) | 3.29 (0.66,16.42) | | 3.17 (2.89,3.47) | 4.60(3.58,5.90) |
| | Hyper endemic | | | | | 14.9(13,17.07) | 36.10 (22.41,58.14) |
| Random Effect | | | | | | | |
| | Household(Variance (Std. Dev.)) | 33.47(5.79) | | 12.94(3.60) | 1.69(1.30) | | 1.37(1.17) |
| | Cluster (Variance (Std.Dev.)) | | 0.01(0.11) | 0.49(0.7) | | 0.015(0.12) | 0.01(0.12) |
| | Residual | 1652 | 41786.4 | 1643.8 | 61472.7 | 70155.9 | |
| | Log likelihood | -846.2 | -20893.2 | -821.9 | -30736.3 | -35077.9 | -30579.3 |
| | ICC | 14% | 30% | 17% | 23% | 30% | 20% |
| | AIC | 1720.4 | 41796.4 | 1677.8 | 61494.7 | 70165.9 | 61188.5 |
| | Bayesian IC | 1827.9 | 41838.2 | 1808.4 | 61592.6 | 70210.3 | 61322 |

use ITN. The combined model demonstrated that children from households with female head (aOR = 1.14, 95% CI 1.1 to 1.19), having television (aOR = 1.08, 95% CI 1.01 to 17.07), coming from richer (aOR = 1.15, 95% CI 1.03 to 1.29) or richest (aOR = 1.31, 95% CI 1.14 to 1.49) wealth quintile, living in rural areas (aOR = 1.38, 95% CI 1.2 to 1.59) and coming from meso endemic (aOR = 4.6, 95% CI 3.59 to 5.9) or hyper endemic area (aOR = 36.08, 95% CI 22.39 to 58.14) were more likely to use ITN. while children from households with 7 to 9 persons (aOR = 0.66, 95% CI 0.6 to 0.73) or > 9 persons (aOR = 0.39, 95% CI 0.35 to 0.43) were less likely to use ITN. The variability in the school age children's models was higher in the household (1.37(1.17)) while the intraclass correlation was 20%.

## Discussions

This paper explored ITN use in CU5 and children of school age in Nigeria using a nationally representative study. Compared to the NDHS report, our study has established associated factors that influence ITN use in CU5 and children of school age by developing the novel multilevel logistic regression model to identify these factors [32]. Findings revealed that household and community- level factors were significantly associated with ITN use in CU5 and children of school age. Factors associated with use of ITN in CU5 include living in rural areas, meso and hyper endemic areas, poor households and owning a television. Similarly, factors associated with use of ITN in children of school age include living in rural area, meso and hyper endemic areas, female headed households, belonging to richer and richest wealth quintile and having television. These findings provide further direction for the malaria elimination strategy on how to enhance ITN use among CU5 and children of school age for malaria elimination effort.

The prevalence of malaria has been documented to be higher in rural areas and in poor population [33–35]. Findings from this study which revealed increased use of ITN in CU5 living in households in lowest wealth quintiles, rural areas showed the effectiveness of ITN campaign in increasing use of ITN to vulnerable and poor population with higher prevalence of malaria.

Furthermore, the finding demonstrates that effort of the malaria control programme in Nigeria to deploy ITN to areas of core endemicity to reduce prevalence and incidence of malaria in these areas is significant. With higher malaria prevalence reported in the northern region which mostly makes up the hyper endemic areas compared to the southern region [36], we can say that efforts to reduce malaria prevalence through use of ITN are effective and needs to be fortified to achieve greater success.

In addition, Higher ITN use and association seen in the meso and hyper endemic areas of the country also shows that people living in these areas may feel more vulnerable to malaria and hence make consistent efforts to use ITNs. Hence, location with hyper endemicity were seen to have high odds of ITN use [27]. The National Malaria Strategic Plan set target of 80% utilization for vector control intervention has not been determined at sub-national level. At state level, only seven states (Ebonyi, Kano, Benue, Jigawa, Kebbi, Plateau, Adamawa) out of the Nigerian 36 states plus FCT had over 80% ITN utilization among CU5. While only Jigawa state had 80% ITN utilization for children of school age. This result reinforces the need for concerted efforts by relevant stakeholders to increase ITN utilization in states where utilization is low, more especially in states that lack donor funding for ITN campaign.

Children of school age in richer and richest wealth quintile were more likely to use ITN. While previous studies showed that children of school age were less prioritized in ITN use at household level [20]. This study further revealed use to be higher in children of school age

from rich households. Efforts should be made by the malaria programme to increase use of ITN in school children from poor households.

WHO recommends a combination of ITN mass campaign, continuous distribution of ITN through multiple channels and several other intervention strategies to eliminate malaria. In Nigeria, continuous distribution via antenatal care and immunization has been prioritized with pockets of school distribution implemented in the country. Plateau state that has been documented to have high prevalence of malaria in children of school age has about one out of every three school age children sleep under ITN. Considering the reported high prevalence of malaria among children of school age, including school distribution of ITN as one of the keep-up channels for achieving universal coverage in Nigeria could be a possible strategy for increasing use of ITN by school children from poor households. Increasing ITN distribution through schools could also sustain the decline in malaria prevalence recorded over the years as school children are reported to be an asymptomatic reservoir for malaria parasites and are the least prioritized in ITN use [20]. Findings also show that children in households with more household members were less likely to sleep under ITN. This finding further buttresses the need to consider family sizes during ITN distribution, as currently there is a cap on maximum number of nets to be given to a household.

## Study strength

The main strength of this study is the representative sample at state, regional and national level to guide ITN utilization strategy and decision making. In addition, while the DHS report presented ITN use in CU5 and children of school age at national level, this paper presents ITN utilization at sub- national levels.

## Study limitation

On the limitation of the study, information on ITN use in children was not verified. The question on use of ITN by children was asked mothers and caregivers of these children and could be subject to social desirability bias (respondents may want to show in their response that they are taking care of their children by making them sleep under ITNs even if this is not true). In addition, the cross-sectional nature of the study design is a limitation to the study as the cause relationship between use of ITN and the observed predisposing factors could not be established. The study was also not able to measure co-variates such as climatic factors, seasonality etc which influence prevalence of malaria. Finally, similar to studies of this nature, this study was not able to measure all factors that could influence ITN use.

## Conclusion

Our study demonstrates higher use of ITN among CU5 and children of school age living in malaria endemic areas and rural areas. However associated factors differ in CU5 and children of school age. To achieve the national target of ITN utilization in children, the authors recommend concerted efforts to increase ITN use in states where utilization remain low while reviewing further household size during ITN campaign. We recommend ITN distribution via schools should be considered as one of the continuous distribution channels to increase use of ITN among children of school age.

## Supporting information

**S1 File. Supplementary file containing Table 3.** Factors associated with ITN use in CU5 and children of school age.
(DOCX)

## Acknowledgments

The authors wish to thank Cameron Taylor of DHS Programs for reviewing this work. Your valuable contribution is highly appreciated.

## Author Contributions

**Conceptualization:** Chinazo N. Ujuju, Chukwu Okoronkwo, Okefu Oyale Okoko, Adekunle Akerele, Chibundo N. Okorie, Samson Babatunde Adebayo.

**Data curation:** Chinazo N. Ujuju.

**Formal analysis:** Chinazo N. Ujuju, Adekunle Akerele.

**Supervision:** Samson Babatunde Adebayo.

**Writing – original draft:** Chinazo N. Ujuju, Chukwu Okoronkwo, Okefu Oyale Okoko, Adekunle Akerele, Chibundo N. Okorie, Samson Babatunde Adebayo.

**Writing – review & editing:** Chinazo N. Ujuju, Chukwu Okoronkwo, Okefu Oyale Okoko, Adekunle Akerele, Chibundo N. Okorie, Samson Babatunde Adebayo.

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
