## [Decision Letter · Decision Letter 0]

4 Apr 2022

PONE-D-22-02975Use of insecticide treated nets in children under five and children of school age in Nigeria: evidence from a secondary data analysis of demographic health surveyPLOS ONE

Dear Dr. Ujuju,

Thank you for submitting your manuscript to PLOS ONE. After careful consideration, we feel that it has merit but does not fully meet PLOS ONE’s publication criteria as it currently stands. Therefore, we invite you to submit a revised version of the manuscript that addresses the points raised during the review process.

We look forward to receiving your revised manuscript.

Kind regards,

Clement Ameh Yaro, Ph.D

Academic Editor

PLOS ONE

Journal Requirements:

Additional Editor Comments:

EDITOR’S COMMENTS

Dear Authors,

The manuscript requires major revision. The authors should critically attend all the comments raised by the reviewers. The multivariate analysis needs to be re-examined. Also, the conclusion should be rewritten to capture the findings of the study.

FIRST REVIEWER

This paper examined the usage of insecticide-treated nets in children under five and school-going children in Nigeria using a secondary analysis of demographic and health survey data. Children are important aspect of malaria elimination program across the world and this paper presents some concerns that the authors need to addressed.

Major comments

1. As the world moves toward elimination, it would bring more valuable information to look into how stratification of malaria prevalence among children at sub-national level would look like under hypo, meso, and hyper endemic regions and how the usages of ITN net were distributed in such stratification.

2. What is the mechanism that could link the sex of the child being female and using ITN in a household? On the other hand, gender of the head of the household is not associated with the ITN use, While sex of the child is of little value for informing policy, I would suggest the authors to use a valid theoretical framework or literature to select variables for the model.

3. The multivariate model showed that malaria endemicity is a predictor for the ITN usage. NCMP usually puts much efforts to bring down the incidence and prevalence in malaria endemic regions, hence this finding suggested that interventions to increase the usage of ITN are working well. Please discuss this point in the paper.

4. The conclusion of “increase availability and access to ITN has resulted … “ is not supported by the findings in the paper. There is no results in the paper showing that the ITN availability or access to ITN increases.

Minor comments

1. The introduction section can be shortened by moving paragraphs between lines 83 and 110 to the methods section.

2. Table 3: put 1 or indicate as “reference” in the reference categories, rather than indicating with a symbol.(less...)

SECOND REVIEWER

Major comments:

1. Why it is called retrospective cross-sectional study?

2. Result session:

There were 32087 U5C in table 1, but slept under ITN (16,671) and not slept (5,769).

There were 54692 children of school age in table 1, but slept under ITN (21,690) and not slept (15,812).

Table 3 should be multivariable logistics regression and use AOR. Need to add reference group.

Please see the table for multivariable logistics regression from other international publication.

3. What is your implication of the study for Nigeria? What is the different information provided by your analysis compared to NDHS findings?

4. Conclusion should provide the specific input to program.

Reviewers' comments:

Reviewer's Responses to Questions

**Comments to the Author**

1. Is the manuscript technically sound, and do the data support the conclusions?

Reviewer #1: No

Reviewer #2: Partly

2. Has the statistical analysis been performed appropriately and rigorously? 

Reviewer #1: Yes

Reviewer #2: No

3. Have the authors made all data underlying the findings in their manuscript fully available?

Reviewer #1: Yes

Reviewer #2: Yes

4. Is the manuscript presented in an intelligible fashion and written in standard English?

Reviewer #1: Yes

Reviewer #2: Yes

5. Review Comments to the Author

Reviewer #1: This paper examined the usage of insecticide-treated nets in children under five and school-going children in Nigeria using a secondary analysis of demographic and health survey data. Children are important aspect of malaria elimination program across the world and this paper presents some concerns that the authors need to addressed.

Major comments

1. As the world moves toward elimination, it would bring more valuable information to look into how stratification of malaria prevalence among children at sub-national level would look like under hypo, meso, and hyper endemic regions and how the usages of ITN net were distributed in such stratification.

2. What is the mechanism that could link the sex of the child being female and using ITN in a household? On the other hand, gender of the head of the household is not associated with the ITN use, While sex of the child is of little value for informing policy, I would suggest the authors to use a valid theoretical framework or literature to select variables for the model.

3. The multivariate model showed that malaria endemicity is a predictor for the ITN usage. NCMP usually puts much efforts to bring down the incidence and prevalence in malaria endemic regions, hence this finding suggested that interventions to increase the usage of ITN are working well. Please discuss this point in the paper.

4. The conclusion of “increase availability and access to ITN has resulted … “ is not supported by the findings in the paper. There is no results in the paper showing that the ITN availability or access to ITN increases.

Minor comments

1. The introduction section can be shortened by moving paragraphs between lines 83 and 110 to the methods section.

2. Table 3: put 1 or indicate as “reference” in the reference categories, rather than indicating with a symbol.

Reviewer #2: Reviewer’s Comments

Major comments:

1. Why it is called retrospective cross-sectional study?

2. Result session:

There were 32087 U5C in table 1, but slept under ITN (16,671) and not slept (5,769).

There were 54692 children of school age in table 1, but slept under ITN (21,690) and not slept (15,812).

Table 3 should be multivariable logistics regression and use AOR. Need to add reference group.

Please see the table for multivariable logistics regression from other international publication.

3. What is your implication of the study for Nigeria? What is the different information provided by your analysis compared to NDHS findings?

4. Conclusion should provide the specific input to program.

6. PLOS authors have the option to publish the peer review history of their article (what does this mean?). If published, this will include your full peer review and any attached files.

Reviewer #1: No

Reviewer #2: **Yes: **Thae Maung Maung

---

## [Author Response · Author response to Decision Letter 0]

21 Apr 2022

21st April 2022

Manuscript Reference#: PONE-D-22-02975

Title: Use of insecticide treated nets in children under five and children of school age in Nigeria: evidence from a secondary data analysis of demographic health survey

RE: Revised manuscript submission and response to reviewer’s comments

Dear Editor, 

This letter is in reference to your email dated April 5, 2022, with reviewers’ comments. We are very much delighted that the manuscript is potentially acceptable for publication in PLOS ONE once the revisions are made. 

We appreciate the reviewers’ insightful comments which significantly improved the manuscript. In the current form the manuscript would make valuable contribution to the literature on this increasingly important topic. 

Please find for your kind consideration the following:

1) A section-by-section response to the comments and suggestions of the reviewers (below). 

2) The revised manuscript, provided as a marked-up copy and a clean copy. 

We hope the changes made on the manuscript meet with your favourable consideration. Please do not hesitate to get in touch if you require any further information. 

Chinazo Ujuju 

Corresponding Author 

Response to Reports

Reviewer #1:

Major comments

 1. As the world moves toward elimination, it would bring more valuable information to look into how stratification of malaria prevalence among children at sub-national level would look like under hypo, meso, and hyper endemic regions and how the usages of ITN net were distributed in such stratification.

Response: Thank you very much for this vital comment. The manuscript presents usage of ITN at sub national level both by geopolitical zone and state level. The use of ITN in hypo, meso and hyper endemic areas is also presented.

 2. What is the mechanism that could link the sex of the child being female and using ITN in a household? On the other hand, gender of the head of the household is not associated with the ITN use, While sex of the child is of little value for informing policy, I would suggest the authors to use a valid theoretical framework or literature to select variables for the model. 

Response: Previous studies included sex as a key socio-demographic factor for health-related outcomes. Similarly, we selected sex of household head and child to examine the association with ITN use. As suggested, literatures that were considered in selecting the variables has been indicated. 

The following literatures included sex as one of the determinants of ITN use in children: 

• Olapeju B, Choiriyyah I, Lynch M, Acosta A, Blaufuss S, Filemyr E, et al. Age and gender trends in insecticide-treated net use in sub-Saharan Africa: a multi-country analysis. Malar J [Internet]. 2018 Dec [cited 2021 Nov 10];17(1):423. Available from: https://malariajournal.biomedcentral.com/articles/10.1186/s12936-018-2575-z

• Nkoka O, Chipeta MS, Chuang Y-C, Fergus D, Chuang K-Y. A comparative study of the prevalence of and factors associated with insecticide-treated nets usage among children under 5 years of age in households that already own nets in Malawi. Malaria Journal [Internet]. 2019 Dec [cited 2021 Oct 8];18(1):43. Available from: https://malariajournal.biomedcentral.com/articles/10.1186/s12936-019-2667-4

• Mensah EA, Anto F. Individual and Community Factors Associated with Household Insecticide-Treated Bednet Usage in the Sunyani West District of Ghana Two Years after Mass Distribution. Journal of Environmental and Public Health. 2020;2020: 1–7. doi:10.1155/2020/7054383

3. The multivariate model showed that malaria endemicity is a predictor for the ITN usage. NCMP usually puts much efforts to bring down the incidence and prevalence in malaria endemic regions, hence this finding suggested that interventions to increase the usage of ITN are working well. Please discuss this point in the paper

Response: Thank you very much for this vital comment. This has been addressed in the manuscript.

 4. The conclusion of “increase availability and access to ITN has resulted … “ is not supported by the findings in the paper. There is no results in the paper showing that the ITN availability or access to ITN increases.

Response: The conclusion has been revised accordingly Thank you.

Minor comments

 1. The introduction section can be shortened by moving paragraphs between lines 83 and 110 to the methods section.

Response: The introduction section has been shortened Thank you.

 2. Table 3: put 1 or indicate as “reference” in the reference categories, rather than indicating with a symbol.(less...)

Response: Table has been revised accordingly Thank you.

SECOND REVIEWER

Major comments:

1. Why it is called retrospective cross-sectional study?

Response: The demographic health survey was conducted in 2018 while secondary analysis for the manuscript was done in 2021. We adjudged the study to be retrospective. However, we have removed “retrospective” for clarity.

2. Result session:

There were 32087 U5C in table 1, but slept under ITN (16,671) and not slept (5,769).

Response: Table 2 was based on children in household who own at least 1 ITN. This variable has been added to Table 1 and also shows number of children in household without ITN for clarity Thank you.

3. There were 54692 children of school age in table 1, but slept under ITN (21,690) and not slept (15,812). 

Response: Table 2 was based on children in households who own at least 1 ITN. This variable has been added to Table 1 and also show number of children in household without ITN.

Table 3 should be multivariable logistics regression and use AOR. Need to add reference group. 

Response: Reference group has been added.

Please see the table for multivariable logistics regression from other international publication. 

Response: Table has been revised accordingly Thank you.

3. What is your implication of the study for Nigeria? What is the different information provided by your analysis compared to NDHS findings? 

Response: Response has been included in the manuscript Thank you.

4. Conclusion should provide the specific input to program

Response: Conclusion revised.

---

## [Decision Letter · Decision Letter 1]

23 Jun 2022

PONE-D-22-02975R1Use of insecticide treated nets in children under five and children of school age in Nigeria: evidence from a secondary data analysis of demographic health surveyPLOS ONE

Dear Dr. Ujuju,

Thank you for submitting your manuscript to PLOS ONE. After careful consideration, we feel that it has merit but does not fully meet PLOS ONE’s publication criteria as it currently stands. Therefore, we invite you to submit a revised version of the manuscript that addresses the points raised during the review process.

We look forward to receiving your revised manuscript.

Kind regards,

Clement Ameh Yaro, Ph.D

Academic Editor

PLOS ONE

Additional Editor Comments:

Manuscript Number: PONE-D-22-02975R1

Title: Use of insecticide treated nets in children under five and children of school age in Nigeria: evidence from a secondary data analysis of demographic health survey.

EDITOR’S COMMENTS

Dear Authors,

The manuscript requires major revision. The authors should attend to all the comments raised by the reviewers. The comments are very important and shouldn’t be neglected if the manuscript should be considered for publication.

REVIEWER 1

The authors have addressed my comments and concerns well and adequately. No further comments from my end.

REVIEWER 2

Major comments:

I would like to suggest to reconsider the analysis because your study objective is to determine the use of ITN in under five and children of school age in Nigeria.

In the result, you mentioned that “more than 80% of CU5 and school children used ITN”. This finding magnify your result of use of ITN because you used the denominator of those who had at least 1 ITN. It cannot be reflect the whole study population of CU5 and school children in Nigeria.

You need to discuss about the % of access to ITN, and use of ITN in total population, and use of ITN in those who had access. Need to compare 2 use of ITN and consider the implications of the study finding.

Therefore, your study objective should focus the access and use of ITN. You cannot omit “access” indicator to improve the ITN utilization implementation in Nigeria.

Minor comments:

1. Why it is called retrospective cross-sectional study?

Response: The demographic health survey was conducted in 2018 while secondary analysis for the manuscript was done in 2021. We adjudged the study to be retrospective. However, we have removed “retrospective” for clarity.

R1: Please remove “retrospective” for the clarity. Here is the sample that used in other manuscript.

“This study analysed the secondary data from MDHS 2015-16, which is a cross-sectional study.”

2. Result session:

There were 32087 U5C in table 1, but slept under ITN (16,671) and not slept (5,769).

Response: Table 2 was based on children in household who own at least 1 ITN. This variable has been added to Table 1 and also shows number of children in household without ITN for clarity Thank you.

R1: Thank you for clarification. Please add total N in the table 1 and 2. I would like to suggest to remove the “Never slept” column for clear vision and eyeballing of the table.

In the tables, Pvalue should be <0.001. No need to describe <0.0001.

3. There were 54692 children of school age in table 1, but slept under ITN (21,690) and not slept (15,812).

Response: Table 2 was based on children in households who own at least 1 ITN. This variable has been added to Table 1 and also show number of children in household without ITN.

R1: OK.

Table 3 should be multivariable logistics regression and use AOR. Need to add reference group.

Response: Reference group has been added.

Please see the table for multivariable logistics regression from other international publication.

Response: Table has been revised accordingly Thank you.

R1: Please mention multivariable logistic regression if you have adjusted the other factors. Please describe AOR instead of OR in the table 3 and in text.* is not the reference value. It is the reference group.

Generally, we describe “1” or Ref in the table. See in the below figure.

3. What is your implication of the study for Nigeria? What is the different information provided by your analysis compared to NDHS findings?

Response: Response has been included in the manuscript Thank you.

4. Conclusion should provide the specific input to program

Response: Conclusion revised.

R1: Please mention study strength and limitation instead of study limitation on line 301. It is mixed for strength and limitation.

REVIEWER 3

The study examined the use of insecticide treated nets and its correlates among children under five and children of school age in Nigeria using the 2018 Nigeria demographic and health survey data. They attempt to address very important public health challenge, especially in sub-Sahara African countries like Nigeria. Their decision to consider both the under 5s and those aged 5-14 is commendable. Studies of this nature are relevant to policymakers and other stakeholder for informed decision making. However, I have a reservation about ignoring the hierarchical structure of the DHS data used in their binary logistic regression models presented.

The background to the study looks good.

Like any other DHS data, the Nigeria DHS data is hierarchical in nature where we have children nested within households, and household nested within clusters (i.e., communities) but the authors did not explain how they account for the hierarchical structure of the data used in this study. Assuming this was not explored during their modelling stage using multilevel (i.e., mixed effect) regression analysis, it could lead to spurious statistical significance with its associated misleading interpretations. Fortunately, we have statistical software packages that allow easy implementation of the multilevel binary logistic regression analysis. Authors are encouraged to explore this and compare the results with the single level binary regression to improve the quality of their results in the manuscript.

Also, since the authors considered both children <5s and 5-14 years and analysed the data separately for these two (2) groups, it could be very useful to pool (i.e., combined) this data and conduct another analysis to assess the correlates for the pooled dataset. This result can then to compared to the separate analysis to inform policy decisions.

Furthermore, the reference categories, the adjusted odds ratios, and p-values in Table 3 are not properly presented, and they should present them in a standard format. The authors should see Table 4 in the publication

https://journals.plos.org/plosone/article?id=10.1371/journal.pone.0257944 for guidance. Also, is not acceptable to present a p-value=0.000 in a manuscript. This should be written as <0.01 or <0.001 in case of p-value=0.0000.

In the conclusion, I struggled to link the analysis done by the authors to their conclusion that “Based on the findings of this study, increased availability and access to ITN has resulted in an increased use by children in rural areas and poor households”. Clearly, this cannot be supported with the data available. I have noted that the previous Reviewer #1 raised same query in item 4 which the authors stated that they have revised in the conclusion and yet they actually did not revise this. I encourage the authors to pay attention to all queries raised and address them to the best of their ability.

The manuscript will also benefit from some few proofreading to improve the message.

REVIEWER 4

Comments:

1. How collinearity test was conducted?

2. How were the variables selected in the final model?

3. Did the authors adjust for the survey design and cluster effect? Please explain.

4. In table 3, both crude and adjusted odds ratio should be shown.

Reviewers' comments:

Reviewer's Responses to Questions

**Comments to the Author**

1. If the authors have adequately addressed your comments raised in a previous round of review and you feel that this manuscript is now acceptable for publication, you may indicate that here to bypass the “Comments to the Author” section, enter your conflict of interest statement in the “Confidential to Editor” section, and submit your "Accept" recommendation.

Reviewer #1: All comments have been addressed

Reviewer #2: (No Response)

Reviewer #3: (No Response)

Reviewer #4: (No Response)

2. Is the manuscript technically sound, and do the data support the conclusions?

Reviewer #1: Yes

Reviewer #2: Partly

Reviewer #3: Partly

Reviewer #4: No

3. Has the statistical analysis been performed appropriately and rigorously? 

Reviewer #1: Yes

Reviewer #2: No

Reviewer #3: Yes

Reviewer #4: No

4. Have the authors made all data underlying the findings in their manuscript fully available?

Reviewer #1: Yes

Reviewer #2: Yes

Reviewer #3: Yes

Reviewer #4: No

5. Is the manuscript presented in an intelligible fashion and written in standard English?

Reviewer #1: Yes

Reviewer #2: Yes

Reviewer #3: Yes

Reviewer #4: Yes

6. Review Comments to the Author

Reviewer #1: The authors have addressed my comments and concerns well and adequately. No further comments from my end.

Reviewer #2: Major comments:

I would like to suggest to reconsider the analysis because your study objective is to determine the use of ITN in under five and children of school age in Nigeria.

In the result, you mentioned that “more than 80% of CU5 and school children used ITN”. This finding magnify your result of use of ITN because you used the denominator of those who had at least 1 ITN. It cannot be reflect the whole study population of CU5 and school children in Nigeria.

You need to discuss about the % of access to ITN, and use of ITN in total population, and use of ITN in those who had access. Need to compare 2 use of ITN and consider the implications of the study finding.

Therefore, your study objective should focus the access and use of ITN. You cannot omit “access” indicator to improve the ITN utilization implementation in Nigeria.

Minor comments:

1. Why it is called retrospective cross-sectional study?

Response: The demographic health survey was conducted in 2018 while secondary analysis for the manuscript was done in 2021. We adjudged the study to be retrospective. However, we have removed “retrospective” for clarity.

R1: Please remove “retrospective” for the clarity. Here is the sample that used in other manuscript.

“This study analysed the secondary data from MDHS 2015-16, which is a cross-sectional study.”

2. Result session:

There were 32087 U5C in table 1, but slept under ITN (16,671) and not slept (5,769).

Response: Table 2 was based on children in household who own at least 1 ITN. This variable has been added to Table 1 and also shows number of children in household without ITN for clarity Thank you.

R1: Thank you for clarification. Please add total N in the table 1 and 2. I would like to suggest to remove the “Never slept” column for clear vision and eyeballing of the table.

In the tables, Pvalue should be <0.001. No need to describe <0.0001.

3. There were 54692 children of school age in table 1, but slept under ITN (21,690) and not slept (15,812).

Response: Table 2 was based on children in households who own at least 1 ITN. This variable has been added to Table 1 and also show number of children in household without ITN.

R1: OK.

Table 3 should be multivariable logistics regression and use AOR. Need to add reference group.

Response: Reference group has been added.

Please see the table for multivariable logistics regression from other international publication.

Response: Table has been revised accordingly Thank you.

R1: Please mention multivariable logistic regression if you have adjusted the other factors. Please describe AOR instead of OR in the table 3 and in text.* is not the reference value. It is the reference group.

Generally, we describe “1” or Ref in the table. See in the below figure.

3. What is your implication of the study for Nigeria? What is the different information provided by your analysis compared to NDHS findings?

Response: Response has been included in the manuscript Thank you.

4. Conclusion should provide the specific input to program

Response: Conclusion revised.

R1: Please mention study strength and limitation instead of study limitation on line 301. It is mixed for strength and limitation.

Reviewer #3: The study examined the use of insecticide treated nets and its correlates among children under five and children of school age in Nigeria using the 2018 Nigeria demographic and health survey data. They attempt to address very important public health challenge, especially in sub-Sahara African countries like Nigeria. Their decision to consider both the under 5s and those aged 5-14 is commendable. Studies of this nature are relevant to policymakers and other stakeholder for informed decision making. However, I have a reservation about ignoring the hierarchical structure of the DHS data used in their binary logistic regression models presented.

The background to the study looks good.

Like any other DHS data, the Nigeria DHS data is hierarchical in nature where we have children nested within households, and household nested within clusters (i.e., communities) but the authors did not explain how they account for the hierarchical structure of the data used in this study. Assuming this was not explored during their modelling stage using multilevel (i.e., mixed effect) regression analysis, it could lead to spurious statistical significance with its associated misleading interpretations. Fortunately, we have statistical software packages that allow easy implementation of the multilevel binary logistic regression analysis. Authors are encouraged to explore this and compare the results with the single level binary regression to improve the quality of their results in the manuscript.

Also, since the authors considered both children <5s and 5-14 years and analysed the data separately for these two (2) groups, it could be very useful to pool (i.e., combined) this data and conduct another analysis to assess the correlates for the pooled dataset. This result can then to compared to the separate analysis to inform policy decisions.

Furthermore, the reference categories, the adjusted odds ratios, and p-values in Table 3 are not properly presented, and they should present them in a standard format. The authors should see Table 4 in the publication https://journals.plos.org/plosone/article?id=10.1371/journal.pone.0257944 for guidance. Also, is not acceptable to present a p-value=0.000 in a manuscript. This should be written as <0.01 or <0.001 in case of p-value=0.0000.

In the conclusion, I struggled to link the analysis done by the authors to their conclusion that “Based on the findings of this study, increased availability and access to ITN has resulted in an increased use by children in rural areas and poor households”. Clearly, this cannot be supported with the data available. I have noted that the previous Reviewer #1 raised same query in item 4 which the authors stated that they have revised in the conclusion and yet they actually did not revise this. I encourage the authors to pay attention to all queries raised and address them to the best of their ability.

The manuscript will also benefit from some few proofreading to improve the message.

Reviewer #4: Comments:

1. How collinearity test was conducted?

2. How were the variables selected in the final model?

3. Did the authors adjust for the survey design and cluster effect? Please explain.

4. In table 3, both crude and adjusted odds ratio should be shown.

7. PLOS authors have the option to publish the peer review history of their article (what does this mean?). If published, this will include your full peer review and any attached files.

Reviewer #1: No

Reviewer #2: **Yes: **Thae Maung Maung

Reviewer #3: **Yes: **Justice Moses Aheto

Reviewer #4: No

---

## [Author Response · Author response to Decision Letter 1]

6 Aug 2022

REVIEWER 1

The authors have addressed my comments and concerns well and adequately. No further comments from my end.

Noted

REVIEWER 2

Major comments:

I would like to suggest to reconsider the analysis because your study objective is to determine the use of ITN in under five and children of school age in Nigeria.

We presume our analysis was done to answer the objectives of the study. The stratification of the analysis into two categories children under five and children of school age meets the objective of our study. Separating the two categories was intentional to ensure that future public health decisions can consider children under five and school aged children independently.

In the result, you mentioned that “more than 80% of CU5 and school children used ITN”. This finding magnify your result of use of ITN because you used the denominator of those who had at least 1 ITN. It cannot be reflect the whole study population of CU5 and school children in Nigeria. 

Thank you for pointing out this very crucial mis-representation of the findings. The population of children under five and children of school age who slept under ITN was 74.3% and 57.8%% respectively and has been included in the result. The result also presented ITN use at state level and described states that had more than 80% ITN coverage in children under five and children above five ( as you presented above). NMEP set vector control utilization target of 80% and this has only been measured at national level using DHS. This manuscript presented state level analysis and the result described states that reached the 80% utilization target for ITN. 

You need to discuss about the % of access to ITN, and use of ITN in total population, and use of ITN in those who had access. Need to compare 2 use of ITN and consider the implications of the study finding. Therefore, your study objective should focus the access and use of ITN. You cannot omit “access” indicator to improve the ITN utilization implementation in Nigeria.

Our study population is children under five and children of school age. Access to ITN is a standard malaria indicator determined at household level. Access to ITN is calculated by dividing the sum of all potential ITN users in the sample by the total number of individuals who spent the previous night in surveyed households. This is based on the assumption that 2 people can sleep under one ITN. As such a household with 4 residents will require 2 ITN. (Reference Kilian, A, H. Koenker, and L. Paintain. 2013. "Estimating population access to insecticide-treated nets from administrative data: correction factor is needed." Malaria journal 12(1): 259. https://malariajournal.biomedcentral.com/articles/10.1186/1475-2875-12-259). As the study did not focus on the entire household but on children under five and children of school age, determining household access to ITN is beyond the scope of this study. 

To determine use of ITN in the study population which is children under five and children of school age, it is not appropriate to ask whether a child living in a household that do not have an ITN slept under ITN. As such the study explored use of ITN in households that have at least one ITN. That shows the child had access to ITN. This is a standard way of determining use of ITN in DHS analysis, presented in the DHS report and is available in several literatures. In addition, our literature already documents efforts by the Nigerian government to increase access. We choose to streamline the focus of this analysis such that our public health message will be clear which is the use of ITN. We have revised the manuscript further to have the introduction and discussion focus on utilization of ITN.

Minor comments:

1. Why it is called retrospective cross-sectional study?

R1: Please remove “retrospective” for the clarity. Here is the sample that used in other manuscript.

“This study analysed the secondary data from MDHS 2015-16, which is a cross-sectional study.”

Retrospective has been removed 

2. Result session:

There were 32087 U5C in table 1, but slept under ITN (16,671) and not slept (5,769).

R1: Thank you for clarification. Please add total N in the table 1 and 2. I would like to suggest to remove the “Never slept” column for clear vision and eyeballing of the table.

In the tables, P value should be <0.001. No need to describe <0.0001.

Total for table 1 and 2 added, P value revised and column on never slept removed from table 2

3. There were 54692 children of school age in table 1, but slept under ITN (21,690) and not slept (15,812).

R1: OK.

Table 3 should be multivariable logistics regression and use AOR. Need to add reference group.

Reference group has been added.

Please see the table for multivariable logistics regression from other international publication.

Table has been revised accordingly 

R1: Please mention multivariable logistic regression if you have adjusted the other factors. Please describe AOR instead of OR in the table 3 and in text.* is not the reference value. It is the reference group.

Generally, we describe “1” or Ref in the table. See in the below figure.

3. What is your implication of the study for Nigeria? What is the different information provided by your analysis compared to NDHS findings?

Response has been included under strength of the manuscript 

4. Conclusion should provide the specific input to program

Conclusion has specific input to program and recommendation

R1: Please mention study strength and limitation instead of study limitation on line 301. It is mixed for strength and limitation.

Study strength and limitation mentioned as recommended

REVIEWER 3

The study examined the use of insecticide treated nets and its correlates among children under five and children of school age in Nigeria using the 2018 Nigeria demographic and health survey data. They attempt to address very important public health challenge, especially in sub-Sahara African countries like Nigeria. Their decision to consider both the under 5s and those aged 5-14 is commendable. Studies of this nature are relevant to policymakers and other stakeholder for informed decision making. However, I have a reservation about ignoring the hierarchical structure of the DHS data used in their binary logistic regression models presented.

The background to the study looks good.

Like any other DHS data, the Nigeria DHS data is hierarchical in nature where we have children nested within households, and household nested within clusters (i.e., communities) but the authors did not explain how they account for the hierarchical structure of the data used in this study. Assuming this was not explored during their modelling stage using multilevel (i.e., mixed effect) regression analysis, it could lead to spurious statistical significance with its associated misleading interpretations. Fortunately, we have statistical software packages that allow easy implementation of the multilevel binary logistic regression analysis. Authors are encouraged to explore this and compare the results with the single level binary regression to improve the quality of their results in the manuscript.

We have considered the suggestion and multilevel logistic regression analysis has been implemented. It’s Important to mention that the study has benefited from implementing a multilevel regression with the models defined at different levels of analysis.

Also, since the authors considered both children <5s and 5-14 years and analysed the data separately for these two (2) groups, it could be very useful to pool (i.e., combined) this data and conduct another analysis to assess the correlates for the pooled dataset. This result can then to compared to the separate analysis to inform policy decisions.

We didn’t think the pooled analysis will be of much public health importance. Children under 5 are critical age in public health and school age children are also significant especially with the increased prevalence of malaria reported in the age group. We are not aware of any significance analysis of data for under 14 children can add to public health. The suggestion is noted and will be considered in future studies. 

Furthermore, the reference categories, the adjusted odds ratios, and p-values in Table 3 are not properly presented, and they should present them in a standard format. The authors should see Table 4 in the publication

https://journals.plos.org/plosone/article?id=10.1371/journal.pone.0257944 for guidance. Also, is not acceptable to present a p-value=0.000 in a manuscript. This should be written as <0.01 or <0.001 in case of p-value=0.0000.

This suggestion is noted and table 3 has been revised

In the conclusion, I struggled to link the analysis done by the authors to their conclusion that “Based on the findings of this study, increased availability and access to ITN has resulted in an increased use by children in rural areas and poor households”. Clearly, this cannot be supported with the data available. I have noted that the previous Reviewer #1 raised same query in item 4 which the authors stated that they have revised in the conclusion and yet they actually did not revise this. I encourage the authors to pay attention to all queries raised and address them to the best of their ability.

Apologies for the oversight. The conclusion has been revised 

The manuscript will also benefit from some few proofreading to improve the message.

Manuscript has been reviewed and errors addressed

REVIEWER 4

Comments:

1. How collinearity test was conducted?

Variance inflation factor has been calculated to determine the extent of collinearity. Multilevel logistic regression analysis was used which ensures that correlation effect was reduced in the estimation of model parameters. 

2. How were the variables selected in the final model?

The variables were selected based on association in the bivariate analysis while the model applied the forward method in variable inclusion in the multilevel logistic regression.

3. Did the authors adjust for the survey design and cluster effect? Please explain.

The design of the DHS survey has adjusted for this effect. Also, the use of multilevel regression analysis ensures that the nested structure of DHS data is taking into effect in the analysis and the variability of the data at different levels of sample selections was measure and presented in table 3. 

4. In table 3, both crude and adjusted odds ratio should be shown.

Given that we have now conducted multilevel logic regression, stratified the model and considered different levels of sampling in the analysis we presume that presenting crude odds ratio may not be necessary in the analysis.

---

## [Decision Letter · Decision Letter 2]

24 Aug 2022

Use of insecticide treated nets in children under five and children of school age in Nigeria: evidence from a secondary data analysis of demographic health survey

PONE-D-22-02975R2

Dear Dr. Ujuju,

We’re pleased to inform you that your manuscript has been judged scientifically suitable for publication and will be formally accepted for publication once it meets all outstanding technical requirements.

Kind regards,

Clement Ameh Yaro, Ph.D

Academic Editor

PLOS ONE

Additional Editor Comments (optional):

The manuscript requires minor revision before consideration for publication, the authors should kindly respond to all the comments raised by the reviewers.

REVIEWER 1

The authors did a great job and provided satisfactory revisions to the paper which significantly improved the message in the manuscript.

However, the authors did not discuss the policy relevance of the significant unobserved household and community level residual (random) effects observed in their results. This should be presented briefly at the discussion section. The authors can benefit from the discussion of the publications below to address this issue and support their findings with same papers. I am happy to provide a quick review within a day after the authors address this.

References

1) https://journals.plos.org/plosone/article/authors?id=10.1371/journal.pone.0269066

2) https://onlinelibrary.wiley.com/doi/full/10.1002/hsr2.453

REVIEWER 2

Thanks for addressing my comments. I would still suggest to show the crude odds ratio. If the authors do not want to show them in the main manuscript, please add a supplementary table showing the crude odds ratio. This will help the readers and future audiences specailly who want to conduct a systematic review and meta-analyses.

REVIEWER 3

Welldone on your research. Please see below my comment to improve the quality of your paper.

1) The independent variables are not robust enough. If your aim is to include variables that would likely influence the utilization of ITNs then some very important variables captured in 2018 NDHS should not be left out bearing in mind that promoting ITN use in Nigeria was a mass national campaign which was publicized through various media channels. It would have been expected that more individual, socio-demographic and media variables (found in the NDHS) such as child birth order, preceding birth interval, mother's educational level, father's educational level, listening to radio, watching TV, reading newspaper are included in the analysis.

The chosen independent variables does not offer a good understanding of the factors contributing to ITN in Nigeria. For instance, child sex - does it really matter if the child is male or female to use the provided ITN? Head of household's sex - most households (99%) in Nigeria are headed by men, when ITN was obtained - how does this contribute to their usage?

In addition, wealth index as used in NDHS serves as an indicator of wealth that is consistent with expenditure and income measures. It was represented as a score of household assets via the principle components analysis method (PCA), this means that household ownership of radio and television is included in the wealth index estimation and so should not be a standalone variable. Therefore, given the set of variables used, it does not come as a surprise that rural residence and wealth index were significantly associated with ITN use.

2) Why include school age children (5 - 14)? The ITN campaign was targeted at children under-5 in Nigeria to whom free ITN was given. School age children (5 - 14) were not given free ITN, they would have to purchase the ITN and this can influence their usage especially if the household is poor. Having both groups of children in the same analysis given the difference in access to the ITN would bias the study findings. I recommend using only children under-5 years for this study if reference is to be made to the ITN campaign in Nigeria OR clearly include this disparity in access to ITN between children under-5 and school age children as a study limitation which readers should approach with caution.

3) It is worth stating that the ITN mass campaign in Nigeria might have influenced the result of this study given that this has been largely focused on poor households in rural areas.

4) The cross-sectional nature of the study design is a limitation on the study's ability to establish a causal relationship between the observed predisposing factors and ITN use in Nigeria.

5) As part of the study limitation, it is worth mentioning the effect of residual confounding as a result of unmeasured co-variates such as climatic factors, season of the year, topography, biomes, etc. These factors influence the prevalence of malaria in a region and thus the use of ITNs.

6) Your study revealed an increase in use of ITN among under-5 children living in poor households, rural areas and meso/hyper endemic areas of Nigeria. Please provide a well-balanced discussion around the contribution of these THREE identified factors to ITN use in Nigeria not only on meso/hyper endemic areas.

Reviewers' comments:

Reviewer's Responses to Questions

**Comments to the Author**

1. If the authors have adequately addressed your comments raised in a previous round of review and you feel that this manuscript is now acceptable for publication, you may indicate that here to bypass the “Comments to the Author” section, enter your conflict of interest statement in the “Confidential to Editor” section, and submit your "Accept" recommendation.

Reviewer #3: (No Response)

Reviewer #4: (No Response)

Reviewer #5: (No Response)

2. Is the manuscript technically sound, and do the data support the conclusions?

Reviewer #3: Yes

Reviewer #4: Yes

Reviewer #5: Yes

3. Has the statistical analysis been performed appropriately and rigorously? 

Reviewer #3: Yes

Reviewer #4: Yes

Reviewer #5: Yes

4. Have the authors made all data underlying the findings in their manuscript fully available?

Reviewer #3: Yes

Reviewer #4: Yes

Reviewer #5: Yes

5. Is the manuscript presented in an intelligible fashion and written in standard English?

Reviewer #3: Yes

Reviewer #4: Yes

Reviewer #5: Yes

6. Review Comments to the Author

Reviewer #3: The authors did a great job and provided satisfactory revisions to the paper which significantly improved the message in the manuscript.

However, the authors did not discuss the policy relevance of the significant unobserved household and community level residual (random) effects observed in their results. This should be presented briefly at the discussion section. The authors can benefit from the discussion of the publications below to address this issue and support their findings with same papers. I am happy to provide a quick review within a day after the authors address this.

References

1)https://journals.plos.org/plosone/article/authors?id=10.1371/journal.pone.0269066

2)https://onlinelibrary.wiley.com/doi/full/10.1002/hsr2.453

Reviewer #4: Thanks for addressing my comments. I would still suggest to show the crude odds ratio. If the authors do not want to show them in the main manuscript, please add a supplementary table showing the crude odds ratio. This will help the readers and future audiences specailly who want to conduct a systematic review and meta-analyses.

Reviewer #5: Welldone on your research. Please see below my comment to improve the quality of your paper.

1) The independent variables are not robust enough. If your aim is to include variables that would likely influence the utilization of ITNs then some very important variables captured in 2018 NDHS should not be left out bearing in mind that promoting ITN use in Nigeria was a mass national campaign which was publicized through various media channels. It would have been expected that more individual, socio-demographic and media variables (found in the NDHS) such as child birth order, preceding birth interval, mother's educational level, father's educational level, listening to radio, watching TV, reading newspaper are included in the analysis.

The chosen independent variables does not offer a good understanding of the factors contributing to ITN in Nigeria. For instance, child sex - does it really matter if the child is male or female to use the provided ITN? Head of household's sex - most households (99%) in Nigeria are headed by men, when ITN was obtained - how does this contribute to their usage?

In addition, wealth index as used in NDHS serves as an indicator of wealth that is consistent with expenditure and income measures. It was represented as a score of household assets via the principle components analysis method (PCA), this means that household ownership of radio and television is included in the wealth index estimation and so should not be a standalone variable.

Therefore, given the set of variables used, it does not come as a surprise that rural residence and wealth index were significantly associated with ITN use.

2) Why include school age children (5 - 14)? The ITN campaign was targeted at children under-5 in Nigeria to whom free ITN was given. School age children (5 - 14) were not given free ITN, they would have to purchase the ITN and this can influence their usage especially if the household is poor. Having both groups of children in the same analysis given the difference in access to the ITN would bias the study findings. I recommend using only children under-5 years for this study if reference is to be made to the ITN campaign in Nigeria OR clearly include this disparity in access to ITN between children under-5 and school age children as a study limitation which readers should approach with caution.

3) It is worth stating that the ITN mass campaign in Nigeria might have influenced the result of this study given that this has been largely focused on poor households in rural areas.

4) The cross-sectional nature of the study design is a limitation on the study's ability to establish a causal relationship between the observed predisposing factors and ITN use in Nigeria.

5) As part of the study limitation, it is worth mentioning the effect of residual confounding as a result of unmeasured co-variates such as climatic factors, season of the year, topography, biomes, etc. These factors influence the prevalence of malaria in a region and thus the use of ITNs.

6) Your study revealed an increase in use of ITN among under-5 children living in poor households, rural areas and meso/hyper endemic areas of Nigeria. Please provide a well-balanced discussion around the contribution of these THREE identified factors to ITN use in Nigeria not only on meso/hyper endemic areas.

7. PLOS authors have the option to publish the peer review history of their article (what does this mean?). If published, this will include your full peer review and any attached files.

Reviewer #3: **Yes: **Justice Moses Aheto

Reviewer #4: No

Reviewer #5: No

---

## [Editor Report · Acceptance letter]

19 Sep 2022

PONE-D-22-02975R2 

Use of insecticide treated nets in children under five and children of school age in Nigeria: evidence from a secondary data analysis of demographic health survey 

Dear Dr. Ujuju:

I'm pleased to inform you that your manuscript has been deemed suitable for publication in PLOS ONE. Congratulations! Your manuscript is now with our production department. 

Kind regards, 

on behalf of

Dr. Clement Ameh Yaro 

Academic Editor

PLOS ONE